

# Molecular identification and antibiotic resistance patterns of diverse bacteria associated with shrimp PL nurseries of Bangladesh: suspecting *Acinetobacter venetianus* as future threat

Abdullah Yasin[1,*], Mst. Khadiza Begum[1,*], Md. Mostavi Enan Eshik[1], Nusrat Jahan Punom[1], Shawon Ahmmed[1,2] and Mohammad Shamsur Rahman[1]

[1] Aquatic Animal Health Group, Department of Fisheries, Faculty of Biological Sciences, University of Dhaka, Dhaka, Bangladesh
[2] Brackishwater Station, Bangladesh Fisheries Research Institute (BFRI), Khulna, Bangladesh
* These authors contributed equally to this work.

Corresponding author
Mohammad Shamsur Rahman, shamsur@du.ac.bd

## ABSTRACT

Shrimp aquaculture has been accomplished with breeding and nursing of shrimp in an artificial environment to fulfill the increasing demand of shrimp consumption worldwide. However, the microbial diseases appear as a serious problem in this industry. The study was designed to identify the diverse bacteria from shrimp PL (post-larvae) nurseries and to profile antibiotic resistance patterns. The rearing water (raw seawater, treated and outlet water) and shrimp PL were collected from eight nurseries of south-west Bangladesh. Using selective agar plates, thirty representative isolates were selected for 16S rRNA gene sequencing, antibiotic susceptibility test and MAR index calculation. Representative isolates were identified as *Aeromonas caviae*, *Pseudomonas monteilii*, *Shewanella algae*, *Vibrio alginolyticus*, *V. brasiliensis*, *V. natriegens*, *V. parahaemolyticus*, *V. shilonii*, *V. xuii*, *Zobellella denitrificans* which are Gram-negative, and *Bacillus licheniformis* and *B. pumilus* which are Gram-positive. Notably, six strains identified as *Acinetobacter venetianus* might be a concern of risk for shrimp industry. The antibiotic resistance pattern reveals that the strain YWO8-97 (identified as *P. monteilii*) was resistant to all twelve antibiotics. Ceftazidime was the most powerful antibiotic since most of the studied strains were sensitive against it. The six strains of *A. venetianus* showed multiple antibiotic resistance patterns. MAR index were ranged from 0.08 to 1.0, and values of 26 isolates were more than 0.2 which means prior high exposure to the antibiotics. From the present study, it can be concluded that shrimp PL nurseries in southern part of Bangladesh are getting contaminated with antibiotic resistant pathogenic bacteria.

## INTRODUCTION

Over the past three decades, the shrimp culture in Asia has been increasing rapidly, and became a major global industry that serves the increasing consumer demand for seafood (*FAO, 2006*). Furthermore, this sector has been contributing significantly to socio-economic development in many poor coastal communities of developing countries (*FAO, 2006*). In 2018–19, the total production of shrimp and prawn in Bangladesh was more than 2,39,855 MT (metric ton), of which 30,036 MT frozen shrimp was exported earning USD 347.54 million (*DoF, 2020*). Frozen shrimp and prawn covers around 80% of the total fisheries export earning over the last decade. In fiscal year 2019–20, solely the tiger shrimp, *Penaeus monodon* contributed 67% to the total earning from exported shrimp and prawn (*DoF, 2020*). The giant tiger shrimp is mainly cultured in the coastal districts of Bangladesh including Cox's Bazar, Chattogram, Khulna, Bagerhat, Satkhira and adjacent districts (*Matin et al., 2016*).

At the beginning, the shrimp farming industry of Bangladesh was dependent on the stocking of wild post-larvae (PL). With the intensification of shrimp farming, the demand of shrimp PL has been increased (*Jannat et al., 2017*). In early 2000s, after transportation from hatcheries the PL of 2–3 cm length were stocked directly into the shrimp ponds resulting in high mortality due to poor acclimatization and predation (*Nuruzzaman, 2006*). To overcome the problem associated with the quality and survival rate of PL, some nurseries have been established by the local people of shrimp culture area *viz*. Khulna and Satkhira districts. According to the description of nursery technician, the hatchery-bred shrimp nauplii or zoea collected from Cox's Bazar are nurtured up to PL stage of several days in concrete tanks in well-designed confined area to meet the demand of shrimp farmers. The rearing water of shrimp PL nurseries is being collected from the sea, and being used after water filtration and disinfection.

The shrimp industry is extremely vulnerable to viral and bacterial diseases (*Karim, Uddin & Uddin, 2017*). Although the vast majority of bacteria are beneficial, a few bacteria are pathogenic containing virulent genes. The bacterial diseases like vibriosis and black shell disease introduce a symbolic restraint on the viable shrimp production (*Bachere et al., 1995*). Bacterial species of Vibrionaceae family *viz. Photobacterium phosphorum, P. leiognathi, Vibrio fischeri, V. harveyi, V. splendidus* and *V. vulnificus* affect the hatcheries.Moreover, *Monodon baculovirus* (MBV), external fouling organisms, *V. harveyi, V. anguillarum, V. vulnificus* are found in shrimp eggs, PL, rearing tank water (treated water), marine water source (raw water) and feed (*Artemia* nauplii and microcapsulated feed) (*Vaseeharan & Ramasamy, 2003*). Some of the *Vibrio* species act as opportunistic pathogens or secondary intruders and they can cause total mortality of cultured shrimp (*Nash et al., 1992*). The bacteria have been found in a commercial freshwater prawn hatchery of Chennai, India are mainly Gram-negative and the species were *Aeromonas* spp., *Pseudomonas* spp., *Vibrio* spp., whereas *Bacillus* spp., and non-spore formers (NSF) were the main Gram-positive bacteria (*Kennedy et al., 2006*). Eleven species of *Vibrio* which cause diseases have been recorded from the shrimp farming systems in Asia (*Lavilla-Pitogo, 1995*). A severe disease problem distresses the farmed black tiger

shrimp in India caused by *V. alginolyticus* and *V. harveyi* (*Karunasagar, Otta & Karunasagar, 1997*). *P. indicus* has been infected by luminous *V. harveyi* in the past, and so as *V. splendidus* affects the *P. monodon* culture (*Prayitno & Latchford, 1995*). The presence of *V. harveyi* also distresses the health of *P. monodon* (*Lavilla-Pitogo & de la Pena, 1998*). Early mortality syndrome (EMS) or acute hepatopancreatic necrosis disease (AHPND) caused by *V. parahaemolyticus* has been noticed in *P. orientalis* (*Xu et al., 1994*) and *P. monodon* (*Chanratchakool et al., 1995*; *Eshik et al., 2018*). However, other species of *Vibrio* viz., *V. owensii*, *V. harveyi*, *V. campbellii* and *V. punensis* containing toxin genes are also responsible for AHPND in shrimps (*Kondo et al., 2015*; *Dong et al., 2017*; *Liu et al., 2018*; *Restrepo et al., 2018*; *Muthukrishnan et al., 2019*). *Acinetobacter venetianus* has been reported as a potential pathogen of red leg disease of freshwater cultured whiteleg shrimp very recently in China, and the genus *Acinetobacter* is causing serious problems associated with high mortality rates in aquaculture (*Huang et al., 2020*). Still, there is lack of information on *A. venetianus* as a pathogen in the world shrimp industry.

Thus, proper biosecurity measures should be applied to prevent the diseases in shrimp aquaculture caused by several microorganisms. The implication of biosecurity in shrimp industry defines the practice of specific pathogen exclusion from broodstock, hatcheries, grow out farms, and from the entire area for disease anticipation (*Lightner, 2003*). Although only three hatcheries of Cox's Bazar and Khulna districts have been authorized in 2018–19 for the use of SPF broodstock for PL production (*DoF, 2020*), most of the hatcheries still use wild broodstock.

In addition, antimicrobial resistance (AMR) has become a potential danger to public health, and the indiscriminate use of antibiotics in animals has been recognized as a major issue nowadays. Different types of antibiotics and other antimicrobial agents are used in shrimp production, predominantly in shrimp hatcheries of Bangladesh and elsewhere (*Holmstrom et al., 2003*; *Uddin & Kader, 2006*; *Thuy, Nga & Loan, 2011*; *Shamsuzzaman & Biswas, 2012*; *Ali et al., 2016*; *Chi et al., 2017*; *Hinchliffe, 2019*).

Thus, the objective of this study was to identify diverse bacterial strains from the water and shrimp PL samples from different nurseries of shrimp farming regions of Bangladesh. Moreover, the antibiotic susceptibility and multiple antibiotic resistance profiles were also investigated for the representative bacterial isolates.

## MATERIALS AND METHODS

### Collection of water and post larvae (PL) samples from nurseries

To characterize the bacteria, shrimp PL and three kinds of water samples *viz.* raw seawater, treated (after filtration and disinfection) water and outlet water were collected from eight nurseries of south-west region of Bangladesh (five nurseries from Dacope Upazilla in Khulna district and three from Satkhira Sadar Upazilla in Satkhira district; Fig. 1). The PL were picked from water using scoop net and packed in a zipper bag with little amount of water, and after a while those were dead. From each nursery, we collected about 200–300 PL, and used as pooled samples. For three kinds of water samples, we took about 500 ml of the water in the sterile bottles from each nursery. All the samples were
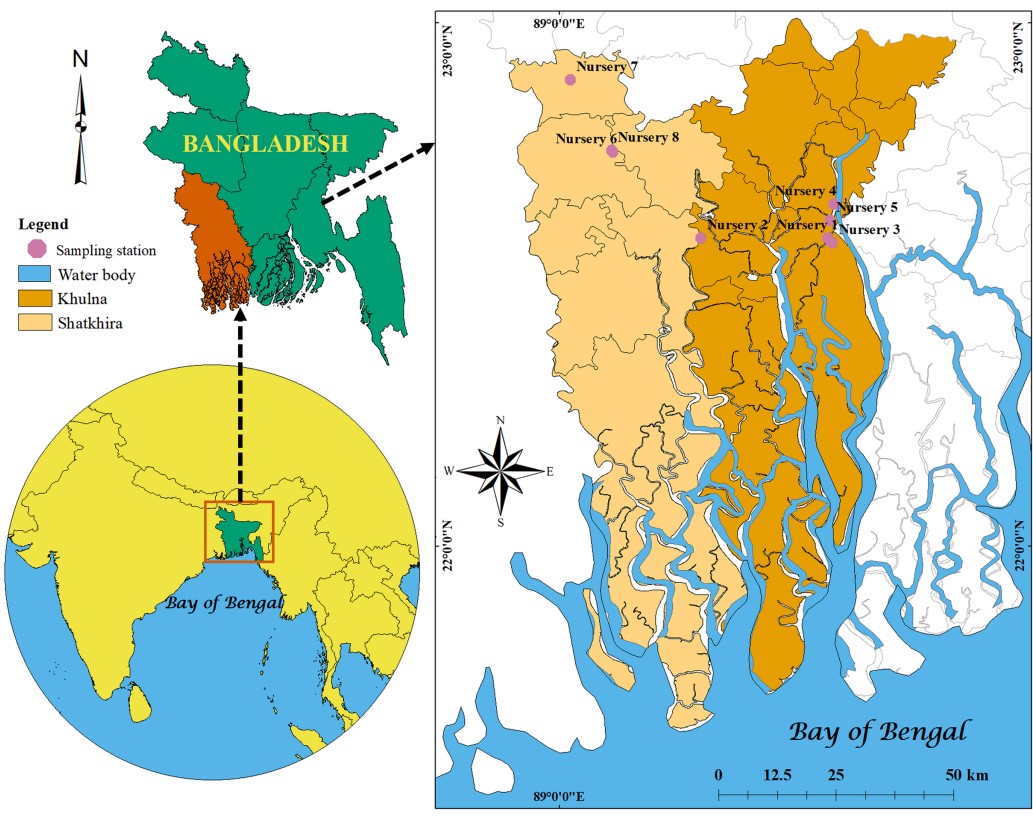

**Figure 1 Locations map.** Map showing the locations of studied shrimp PL nurseries in Khulna and Satkhira districts of Bangladesh created by ArcGIS version 10.8.

transferred in the sample box with ice (*APHA, 1998*) and finally brought to the laboratory for further analysis.

## Sample processing for bacterial culture

PL samples were homogenized with physiological saline using sterile mortar-pestle at 1:9 ratios and taken separately into sterile zipper bags. Then, 100 μL of each sample solution was taken with 900 μL sterile physiological saline separately and diluted up to $10^{-5}$ fold to isolate different bacteria on TCBS (thiosulphate citrate bile salt), EMB (eosin methylene blue), SS (Salmonella-Shigella), and MSA (mannitol salt agar) agar media. For the isolation of *Vibrio* spp. the processed samples were inoculated in alkaline peptone water (APW) and incubated for 6–18 h at 37 °C for enrichment.

## Isolation of diverse bacterial colonies from selective agar plates

From diluted PL samples, raw water and outlet water samples, 100 μL of $10^{-1}$ and $10^{-3}$ dilution were inoculated on sterile TCBS (Oxoid, Basingstoke, UK), SS (Oxoid, Basingstoke, UK), MSA (Oxoid, Basingstoke, UK), and EMB (Oxoid, Basingstoke, UK) agar plates. Only from treated water, 100 μL of processed undiluted water sample and $10^{-2}$ dilution sample was inoculated on different agar plates. Then, the samples were spread thoroughly by using a sterile glass spreader. The plates were transferred into an incubator

at 37 °C for 18–24 h. After incubation, the well discrete colonies from different plates were selected for subculture on TCBS, SS, MSA and EMB agar plates. Overall, 98 colonies were selected primarily and sub-cultured on respective selective agar plates for isolated single colonies. The selected colonies from respective agar plates were taken and streaked on the fresh respective plates to get isolated single colony for pure culture. Based on the colony morphology on selective agar plates, thirty representative isolates were cultured by streaking plate technique on TSA plates for further works.

## Molecular identification of the isolates

The heat extraction method was performed for the DNA extraction from 30 representative bacterial isolates (*Rahman et al., 2014*). The target DNA amplicon was amplified using universal primers for 16S rRNA gene: 27F 5′-AGAGTTTGATCCTGGCTCAG-3′ and 1492R 5′-CGGTTACCTTGTTACGACTT-3′ (*Weisburg et al., 1991*). The polymerase chain reaction (PCR) was done following the reaction mixture and thermal cycling condition as described by *Punom et al. (2016)*. The amplified products were checked for the desired amplicon in 1% agarose gel. The amplified DNA was further purified with the Wizard SV Gel and PCR Clean-Up System (Promega, Madison, WI, USA) according to the manufacturer instructions prior to sequencing. The sequencing of thirty PCR products was performed using the BigDye Terminator v 3.1 Cycle sequencing Kit (Applied Biosystems, Waltham, MA, USA) according to the manufacturer instructions and capillary electrophoresis was done using the ABI Genetic Analyzer (Applied Biosystems, Waltham, MA, USA). To view the DNA sequences, the Geospizas Finch TV version 1.4 was used. BLAST (Basic Local Alignment Search Tool) was used for comparing primary sequence information.

A neighbor-joining phylogenetic tree was constructed using MEGA X software (*Kumar et al., 2018*) for the comparative analysis of 16S rRNA sequences of the bacterial isolates of the present study and the reference sequences deposited in the NCBI database. The tree was drawn to scale, with branch lengths in the same units as those of the evolutionary distances used to infer the phylogenetic tree. The evolutionary distances among these representative isolates were computed using the maximum composite likelihood method.

## Antibiotic susceptibility test

The Kirby-Bauer disc diffusion technique (*Bauer et al., 1966*) was performed to determine the sensitivity or resistance of the isolates against 12 commonly available antibacterial compounds. These were Ampicillin (10 μg), Azithromycine (15 μg), Chloramphenicol (30 μg), Ciprofloxacine (5 μg), Ceftriaxone (30 μg), Erythromycine (15 μg), Gentamycin (10 μg), Sulphamethoxazole-Trimethoprime (25 μg), Trimethoprime (5 μg), Penicillin G (10 μg), Tetracycline (30 μg) and Ceftazidime (30 μg).

Thirty representative isolates were inoculated in Mueller-Hinton Broth (MHB) (Oxoid, UK) and incubated for 24 h and then the broth culture was spread on the surface of the Mueller-Hinton Agar (MHA). The antibiotic discs (Oxoid, Basingstoke, UK) were

applied on the surface of the agar plates and incubated for 24 h at 37 °C. Finally, the zone of inhibition was measured to detect susceptibility of the bacteria (*CLSI, 2013*).

## Multiple antibiotic resistance (MAR) index of the studied isolates

The multidrug resistance and MAR index of the representative isolates were calculated against twelve antibiotics. The bacterial strains resistant against three or more antibiotics were considered as multidrug resistant strains (*Schwarz et al., 2010*). The MAR index values of the bacterial isolates were calculated as: a/b; where 'a' represents the number of antibiotics the strain was resistant to, and 'b' denotes the total number of antibiotics the strain was tested (*Krumperman, 1983*).

# RESULTS

## Colony morphology of the representative bacterial isolates

A total of thirty representative isolates were selected from four selective agar plates for further study on the basis of different colors and sizes. The colony morphologies of representative isolates are presented in Table 1.

## Molecular identification of the representative isolates

Based on the colony morphology, 30 representative bacterial isolates were selected for 16S rRNA gene sequencing in the present study. After amplification, all the representative isolates showed positive band estimated at 1,500 bp in gel electrophoresis (Fig. 2). After sequencing, the identification of 16S rRNA gene sequences of the representative isolates by nucleotide BLAST of NCBI is described in Table 2. Among the isolates from eight nurseries, 15 isolates showed their identity with different strains of *Vibrio* spp. deposited in the NCBI database. YPL1-2 showed 100% identity with *V. xuii* strain DSM 17185. The isolates YPL1-4, YWO1-14, YPL2-19 and YPL4-52 revealed their similarity with *V. shilonii* strain having 100% query coverage and identity. YPL2-16 showed 100% identity and query cover with *V. natriegens*. The 16S rRNA gene sequence of YPL-18 was identified as *V. parahaemolyticus*. After searching the sequences in the GenBank, six isolates (YPL2-20, YWO3-27, YWO3-29, YWT6-68, YWO6-72 and YWT7-80) redirect their 100% identity with *V. alginolyticus*. YWR2-22 and YWT5-55 showed identity more than 99% with *V. brasiliensis*. On the other hand, 15 remaining isolates showed their identity with eight different species. YWO1-12 and YWR2-23 were identified as *Bacillus licheniformis* and *B. pumilus*, respectively. These two isolates are Gram-positive.

Besides, the isolates YPL3-35, YWT4-39, YPL6-75, YWR7-79, YWO7-86 and YWR8-91 showed their 100% similarity with *Acinetobacter venetianus*. It is noteworthy that these isolates were screened mainly on EMB agar plates showing pinkish colored small colonies except YPL6-75, which was isolated from SS agar plate showing cream colored small size colony. The isolates YWO5-61 and YPL5-62 were identified as *Zobellella denitrificans* and *Aeromonas caviae*, respectively. The sequences of YPL5-64, YPL7-89 and YWT8-94 showed 100% query cover and identity with *Shewanella algae*. The sequence of YWO8-97 reflects 100% identity and query cover with *Pseudomonas monteilii*, and

**Table 1 Colony morphology of bacterial isolates.** Colony morphology of representative thirty bacterial isolates from eight different shrimp PL nurseries of Khulna and Satkhira.

| Isolate No. | Sample code | Sampling location | Sample source | Culture media | Culture condition | Colony color | Size |
|---|---|---|---|---|---|---|---|
| YPL1-2 | YPL1/MSA/1 | Khulna | PL | MSA | Direct plating | Pinkish | Small |
| YPL1-4 | YPL1/TCBS/1 | Khulna | PL | TCBS | Direct plating | Greenish | Small |
| YWO1-12 | YWO1/MSA/1 | Khulna | Outlet water | MSA | Direct plating | Whitish | Small |
| YWO1-14 | YWO1/TCBS/6h/1 | Khulna | Outlet water | TCBS | APW enrichment | Greenish | Small |
| YPL2-16 | YPL2/MSA/1 | Khulna | PL | MSA | Direct plating | Whitish red | Small |
| YPL2-18 | YPL2/TCBS/6h/1 | Khulna | PL | TCBS | APW enrichment | Yellowish | Small |
| YPL2-19 | YPL2/ TCBS/6h/2 | Khulna | PL | TCBS | APW enrichment | Greenish | Small |
| YPL2-20 | YPL2/TCBS/Raw | Khulna | PL | TCBS | Direct plating | Greenish yellow | Small |
| YWR2-22 | YWR2/TCBS/Raw/1 | Khulna | Raw sea water | TCBS | Direct plating | Greenish | Large |
| YWR2-23 | YWR2/MSA/1 | Khulna | Raw sea water | MSA | Direct plating | Whitish | Small |
| YWO3-27 | YWO3/SS/1 | Khulna | Outlet water | SS | Direct plating | Pinkish | Large |
| YWO3-29 | YWO3/TCBS/1 | Khulna | Outlet water | TCBS | Direct plating | Yellowish | Large |
| YPL3-35 | YPL3/EMB/1 | Khulna | PL | EMB | Direct plating | Pinkish | Small |
| YWT4-39 | YWT4/EMB/1 | Khulna | Treated water | EMB | Direct plating | Pinkish | Small |
| YPL4-52 | YPL4/TCBS/6h/3 | Khulna | PL | TCBS | APW enrichment | Yellowish | Medium |
| YWT5-55 | YWT5/MSA/2 | Khulna | Treated water | MSA | Direct plating | Yellowish | Large |
| YWO5-61 | YWO5/ SS/1 | Khulna | Outlet water | SS | Direct plating | Colorless | Medium |
| YPL5-62 | YPL5/EMB/1 | Khulna | PL | EMB | Direct plating | Pinkish | Medium |
| YPL5-64 | YPL5/SS/1 | Khulna | PL | SS | Direct plating | Colorless | Medium |
| YWT6-68 | YWT6/MSA/1 | Satkhira | Treated water | MSA | Direct plating | Cream | Small |
| YWO6-72 | YWO6/MSA/1 | Satkhira | Outlet water | MSA | Direct plating | Cream | Large |
| YPL6-75 | YPL6/SS/1 | Satkhira | PL | SS | Direct plating | Cream | Small |
| YWR7-79 | YWR7/EMB/1 | Satkhira | Raw sea water | EMB | Direct plating | Pinkish | Small |
| YWT7-80 | YWT7/TCBS/6h/1 | Satkhira | Treated water | TCBS | APW enrichment | Yellow | Large |
| YWO7-86 | YWO7/EMB/1 | Satkhira | Outlet water | EMB | Direct plating | Pinkish | Small |
| YPL7-89 | YPL7/SS/1 | Satkhira | PL | SS | Direct plating | Pinkish | Small |
| YWR8-91 | YWR8/EMB/1 | Satkhira | Raw sea water | EMB | Direct plating | Pinkish | Small |
| YWT8-94 | YWT8/SS/1 | Satkhira | Treated water | SS | Direct plating | Pink | Small |
| YWO8-97 | YWO8/EMB/1 | Satkhira | Outlet water | EMB | Direct plating | Purple | Small |
| YPL8-98 | YPL8/EMB/2 | Satkhira | PL | EMB | Direct plating | Pinkish | Small |

**Note:**
*Small when diameter is ≤1 mm, Medium when diameter is 1.1–≤2 mm, Large when diameter is >2 mm.

YPL8-98 (not in full length due to noise in chromatogram; only 630 bp were used for blast search) showed similarity with *Pseudomonas* spp. in the NCBI database.

The reported sequences in this study have been deposited in the NCBI GenBank database under accession numbers from MT368010 to MT368038.

## Phylogeny analysis

The phylogenetic tree was constructed based on the neighbor-joining method using the 16S rRNA gene sequences of 29 representative isolates (YPL8-98 was excluded because of relatively short sequence), and other reference sequences downloaded from the

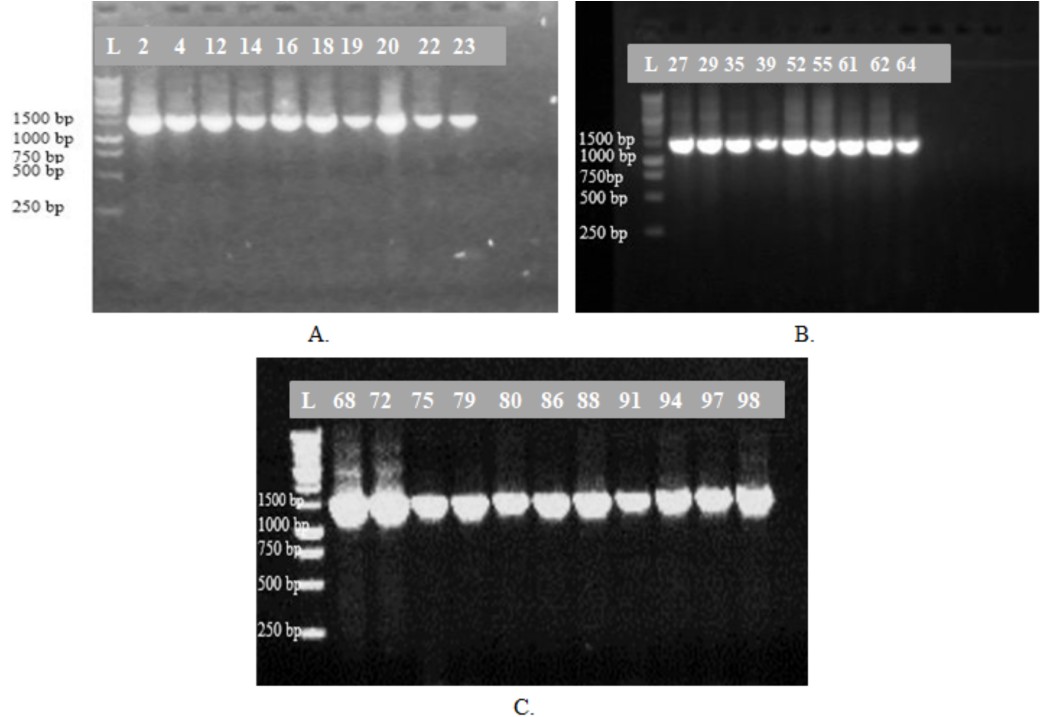

**Figure 2 Gel photographs.** Amplification of 16S rRNA gene of 30 representative bacterial isolates (A)–(C); L-denotes molecular marker and numbers denotes the representative bacterial isolates.

GenBank (Fig. 3). The tree showed the taxonomic status of the studied 29 bacterial isolates through comparing their 16S rRNA gene sequences with the other homologous sequences.

The identified bacterial strains have been summarized in the Table 3 according to the sample types (water and shrimp PL) collected from shrimp PL nurseries.

## Antibiotic susceptibility pattern of studied isolates

Table 4 shows antibiotic susceptibility pattern of 30 representative bacterial isolates. Among the studied isolates, 26 bacterial strains were resistant to three or more antibiotics. The study found a very remarkable isolate (YWO8-97; identified as *P. monteilii*) from outlet water of one nursery of Satkhira district that was resistant to all 12 antibiotics. On the other hand, *V. shilonii*, *V. brasiliensis*, *B. licheniformis*, *V. natriegens*, *V. parahaemolyticus*, *B. pumilus*, *Z. denitrificans* were quite sensitive to the tested antibiotics. In this study, among six isolates identified as *V. alginolyticus*, five isolates were resistant to ampicillin and gentamycin. All isolates of *V. alginolyticus* were resistant to penicillin. Fig. 4 illustrates the percentage of the isolates showing different types of susceptibility pattern against the tested antibiotics. No isolates were sensitive to tetracycline except *V. xuii* (Fig. 4). Against 12 antibiotics, the resistant pattern of 30 studied isolates was as: P > E = W > TE > AMP = CN = SXT > CIP > AZM = C > CAZ > CRO. Only ceftazidime was the most effective antibiotic because 83.3% of the studied isolates was sensitive against it (Fig. 4). The isolates identified as *A. venetianus* showed more

**Table 2 Identification of bacterial isolates.** Identification of thirty representative bacterial isolates by nucleotide BLAST search of NCBI database using 16S rRNA gene sequences.

| Serial no. | Isolate no. (Strain designation) | Identified species | Most relevant strain name | Length of sequences (bp) | Max score | Query cover | E value | Identity match | GenBank accession no. of corresponding sequence | GenBank accession no. of studied strains |
|---|---|---|---|---|---|---|---|---|---|---|
| 1 | YPL1-2 | *Vibrio xuii* | DSM17185 | 1137 | 2100 | 100% | 0.0 | 100% | MH315813.1 | MT368010 |
| 2 | YPL1-4 | *V. shilonii* | VSS-012 | 1426 | 2634 | 100% | 0.0 | 100% | FJ485944.1 | MT368011 |
| 3 | YWO1-12 | *Bacillus licheniformis* | R-QL-77-10 | 1427 | 2636 | 100% | 0.0 | 100% | MT078630.1 | MT368012 |
| 4 | YWO1-14 | *V. shilonii* | VSS-012 | 1422 | 2627 | 100% | 0.0 | 100% | FJ485944.1 | MT368013 |
| 5 | YPL2-16 | *V. natriegens* | P14-2 | 1422 | 2627 | 100% | 0.0 | 100% | KC261284.1 | MT368014 |
| 6 | YPL2-18 | *V. parahaemolyticus* | 19-021-D1 | 1423 | 2628 | 100% | 0.0 | 100% | CP046411.1 | MT368015 |
| 7 | YPL2-19 | *V. shilonii* | VSS-012 | 1425 | 2632 | 100% | 0.0 | 100% | FJ485944.1 | MT368016 |
| 8 | YPL2-20 | *V. alginolyticus* | 5-19 | 1424 | 2630 | 100% | 0.0 | 100% | MN945282.1 | MT368017 |
| 9 | YWR2-22 | *V. brasiliensis* | XSH | 1426 | 2571 | 100% | 0.0 | 99.23% | MT071607.1 | MT368018 |
| 10 | YWR2-23 | *Bacillus pumilus* | 10B2-13 | 1420 | 2623 | 100% | 0.0 | 100% | MK603127.1 | MT368019 |
| 11 | YWO3-27 | *V. alginolyticus* | 5-19 | 1423 | 2628 | 100% | 0.0 | 100% | MN945282.1 | MT368020 |
| 12 | YWO3-29 | *V. alginolyticus* | 5-19 | 1426 | 2634 | 100% | 0.0 | 100% | MN945282.1 | MT368021 |
| 13 | YPL3-35 | *Acinetobacter venetianus* | PFBCI | 1409 | 2603 | 100% | 0.0 | 100% | LN875372.1 | MT368022 |
| 14 | YWT4-39 | *A. venetianus* | ICP1 | 1417 | 2617 | 100% | 0.0 | 100% | MN542884.1 | MT368023 |
| 15 | YPL4-52 | *V. shilonii* | VSS-012 | 1423 | 2628 | 100% | 0.0 | 100% | FJ485944.1 | MT368024 |
| 16 | YWT5-55 | *V. brasiliensis* | IS014 | 1425 | 2569 | 100% | 0.0 | 99.23% | KR186076.1 | MT368025 |
| 17 | YWO5-61 | *Zobellella denitrificans* | F13 | 1417 | 2612 | 100% | 0.0 | 99.93% | CP012621.1 | MT368026 |
| 18 | YPL5-62 | *Aeromonas caviae* | BTNGPSA3 | 1417 | 2617 | 100% | 0.0 | 100% | MK958566.1 | MT368027 |
| 19 | YPL5-64 | *Shewanella algae* | SFH3 | 1422 | 2627 | 100% | 0.0 | 100% | MG738264.1 | MT368028 |
| 20 | YWT6-68 | *V. alginolyticus* | FA2 | 1432 | 2645 | 100% | 0.0 | 100% | CP042449.1 | MT368029 |
| 21 | YWO6-72 | *V. alginolyticus* | 5-19 | 1430 | 2641 | 100% | 0.0 | 100% | MN945282.1 | MT368030 |
| 22 | YPL6-75 | *A. venetianus* | PFBCI | 1375 | 2540 | 100% | 0.0 | 100% | LN875372.1 | MT368031 |
| 23 | YWR7-79 | *A. venetianus* | PFBCI | 1373 | 2536 | 100% | 0.0 | 100% | LN875372.1 | MT368032 |
| 24 | YWT7-80 | *V. alginolyticus* | 5-19 | 1414 | 2612 | 100% | 0.0 | 100% | MN945282.1 | MT368033 |
| 25 | YWO7-86 | *A. venetianus* | PFBCI | 1412 | 2608 | 100% | 0.0 | 100% | LN875372.1 | MT368034 |
| 26 | YPL7-89 | *S. algae* | SFH3 | 1423 | 2628 | 100% | 0.0 | 100% | MG738264.1 | MT368035 |
| 27 | YWR8-91 | *A. venetianus* | ICP1 | 1375 | 2540 | 100% | 0.0 | 100% | MN542884.1 | MT368036 |
| 28 | YWT8-94 | *S. algae* | SFH3 | 1416 | 2615 | 100% | 0.0 | 100% | MG738264.1 | MT368037 |
| 29 | YWO8-97 | *Pseudomonas monteilii* | ER30 | 1412 | 2608 | 100% | 0.0 | 100% | MT124555.1 | MT368038 |
| 30 | YPL8-98 | *Pseudomonas* sp. (not in full length) | YX6 | 630 | 1157 | 99% | 0.0 | 99.84% | KP789459.1 | Not submitted |

resistance pattern as well as *S. algae*. The six isolates identified as *A. venetianus* were resistant to penicillin G, trimethoprim, tetracycline, and consecutively five isolates were resistant to azithromycin, ciprofloxacin and erythromycin. However, no isolate showed resistance against ceftazidime and gentamycin (Fig. 5).

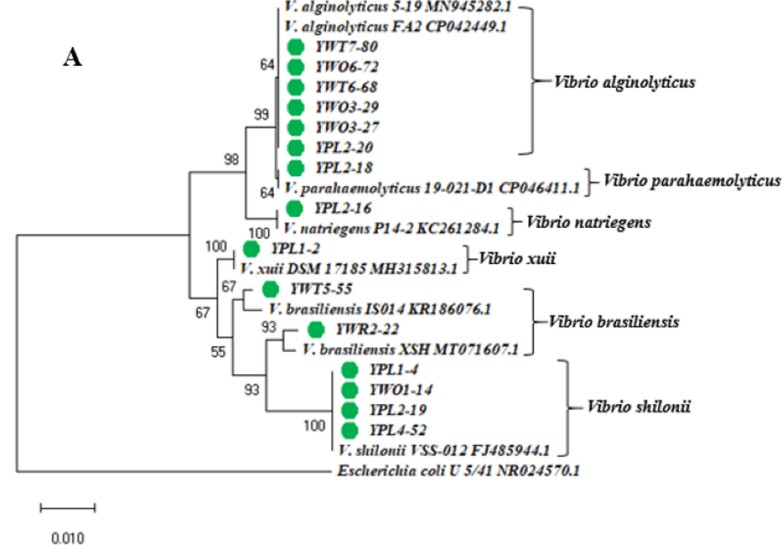

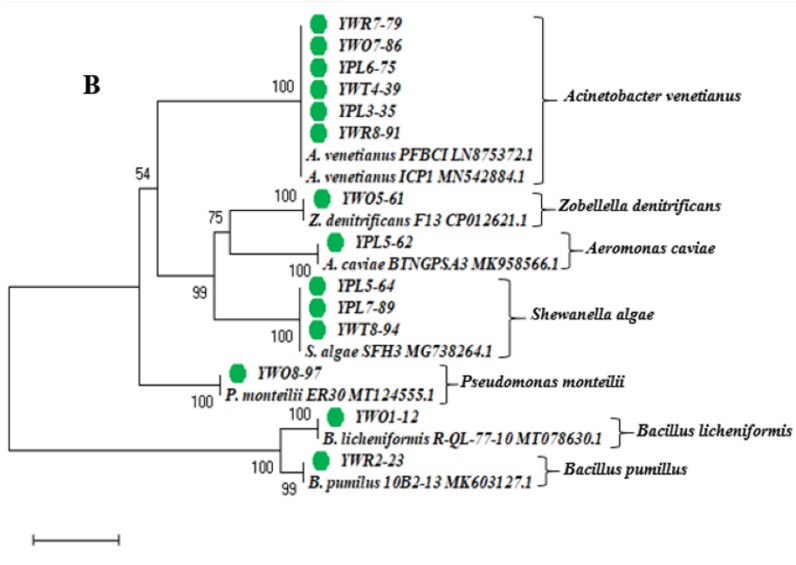

**Figure 3** **Phylogenetic tree.** Neighbor-joining phylogenetic tree constructed using MEGA X based on 16S rRNA gene sequences showing the relationship between (A) *Vibrio* spp.; (B) other different gram negative and gram positive bacterial species. Taxa labeled with green circles indicate the bacterial strains isolated from four types of samples collected from eight different shrimp nurseries in the present study. Other taxa were obtained from the NCBI database.   

## Multiple antibiotic resistance (MAR) index of the studied isolates

Among 30 representative isolates, MAR index values were ranged from 0.08 to 1.0 and values were >0.2 in 26 bacterial strains which means previous high exposure to the antibiotics and thereby causing risk of antibiotic resistance (Fig. 6). Therefore, 86.7% isolates were multidrug resistant including all the six isolates identified as *A. venetianus* having MAR index from 0.5 to 0.75. The studied 14 bacterial strains identified as *Vibrio*

**Table 3 Identified bacterial isolates.** Identified bacterial isolates in four different types of samples collected from eight different shrimp PL nurseries.

| Types of sample | Isolate no. (Identified species) |
| --- | --- |
| Raw seawater | YWR2-22 (*V. brasiliensis*), YWR2-23 (*B. pumillus*), YWR7-79 (*A. venetianus*), YWR8-91 (*A. venetianus*), |
| Treated water | YWT4-39 (*A. venetianus*), YWT5-55 (*V. brasiliensis*), YWT6-68 (*V. alginolyticus*), YWT7-80 (*V. alginolyticus*), YWT8-94 (*S. algae*) |
| Outlet water | YWO1-12 (*B. licheniformis*), YWO1-14 (*V. shilonii*), YWO3-27 (*V. alginolyticus*), YWO3-29 (*V. alginolyticus*), YWO5-61 (*Z. denitrificans*), YWO6-72 (*V. alginolyticus*), YWO7-86 (*A. venetianus*), YWO8-97 (*P. monteilii*) |
| Shrimp PL | YPL1-2 (*V. xuii*), YPL1-4 (*V. shilonii*), YPL2-16 (*V. natriegens*), YPL2-18 (*V. parahaemolyticus*), YPL2-19 (*V. shilonii*), YPL2-20 (*V. alginolyticus*), YPL3-35 (*A. venetianus*), YPL4-52 (*V. shilonii*), YPL5-62 (*A. caviae*), YPL5-64 (*S. algae*), YPL6-75 (*A. venetianus*), YPL7-89 (*S. algae*), YPL8-98 (*Pseudomonas* spp.) |

**Table 4 Antibiotic susceptibility patterns of bacterial isolates.** Antibiotic susceptibility patterns of thirty representative bacterial isolates from raw sea water, treated water, outlet water and PL samples collected from eight different shrimp PL nurseries against 12 antibiotics.

| Isolate No. | Identified species | Antibiotic susceptibility | | |
| --- | --- | --- | --- | --- |
| | | Sensitive | Intermediate | Resistant |
| YPL1-2 | *V. xuii* | AMP, AZM, C, CIP, CRO, CN, TE, CAZ | – | E, SXT, W, P |
| YPL1-4 | *V. shilonii* | AMP, AZM, C, CIP, CRO, CN, P, CAZ | E, TE | SXT, W |
| YWO1-12 | *B. licheniformis* | AMP, CIP, CRO, CN, SXT, W, P | AZM, C | E, TE, CAZ |
| YWO1-14 | *V. shilonii* | AMP, AZM,C, CIP, CN, P, CAZ | TE | CRO,E, SXT,W, |
| YPL2-16 | *V. natriegens* | AMP, AZM, C, CIP, CRO, E, SXT, P, CAZ | – | CN, W, TE |
| YPL2-18 | *V. parahaemolyticus* | AMP, AZM, C, CIP, CRO, E, SXT, W, S | – | CN, P, TE |
| YPL2-19 | *V. shilonii* | AMP, AZM, C, CRO, P, CAZ | CIP, TE | E, CN, SXT, W |
| YPL2-20 | *V. alginolyticus* | AMP, C, CRO, CN, SXT, W, CAZ | AZM, | CIP, E, P, TE |
| YWR2-22 | *V. brasiliensis* | AMP, AZM, C, CIP, CRO, E, SXT, W, P, CAZ | TE | CN |
| YWR2-23 | *B. pumilus* | AMP, AZM, C, CIP, E, CN, SXT, W, P | CRO, TE | CAZ |
| YWO3-27 | *V. alginolyticus* | C, CRO, CAZ | W | AMP, AZM, CIP, E, CN, SXT, P, TE |
| YWO3-29 | *V. alginolyticus* | AZM, C, CRO, SXT, W, CAZ | CIP, E, TE | AMP, CN, P |
| YPL3-35 | *A. venetianus* | AMP, CN, SXT, CAZ | C, CRO | AZM, CIP, E, W, P, TE |
| YWT4-39 | *A. venetianus* | AMP, AZM, C,CIP, CN, SXT, CAZ | CRO, E | W, P, TE |
| YPL4-52 | *V. shilonii* | AMP, C, CN, P,CAZ | AZM, CIP | SXT, W, TE |
| YWT5-55 | *V. brasiliensis* | AMP, AZM, CIP, CRO, E, CN, CAZ | TE | C, SXT, W, P |
| YWO5-61 | *Z. denitrificans* | AMP, AZM, C, CIP, CRO, SXT, W, P, CAZ | E, TE | CN |
| YPL5-62 | *A. caviae* | AZM, C, CRO, SXT, W, CAZ | CIP,E | AMP, CN, P, TE |
| YPL5-64 | *S. algae* | AZM, CRO, CN, CAZ | C, CIP | AMP, E, SXT, W, P, TE |
| YWT6-68 | *V. alginolyticus* | AZM, C, CRO, SXT, W, CAZ | TE | AMP, CIP, E, CN, P |
| YWO6-72 | *V. alginolyticus* | AZM, C, CRO, E, SXT, W, CAZ | CIP, TE | AMP, CN, P |
| YPL6-75 | *A. venetianus* | CN, SXT | CRO | AMP, AZM, C, CIP, E, W, P, TE, CAZ |
| YWR7-79 | *A. venetianus* | CN, SXT, CAZ | CRO | AMP, AZM, C, CIP, E, W, P, TE |
| YWT7-80 | *V. alginolyticus* | C, CRO, SXT, W, CAZ | AZM, TE | AMP, CIP, E, CN, P |
| YWO7-86 | *A. venetianus* | C, CN | CRO, CAZ | AMP, AZM, CIP, E, SXT, W, P, TE |
| YPL7-89 | *S. algae* | AZM, CRO, CAZ | TE | AMP, C, CIP, E, CN, SXT, W, P |
| YWR8-91 | *A. venetianus* | CN, CAZ | C, CRO | AMP, AZM, CIP, E, SXT, W, P, TE |
| YWT8-94 | *S. algae* | AMP, AZM, CRO, P, CAZ | – | C, CIP, E, CN, SXT, W, TE |
| YWO8-97 | *P. monteilii* | – | – | AMP, AZM, C, CIP, CRO, E,CN, AXT, W, P, TE, CAZ |
| YPL8-98 | *Pseudomonas* sp. | AZM, CN, CAZ | CRO | AMP, C, CIP, E, SXT, W, P, TE |

**Note:**
AMP, Ampicillin; AZM, Azithromycin; C, Chloramphenicol; CIP, Ciprofloxacin; CRO, Ceftriaxone; E, Erythromycin; CN, Gentamycin; SXT, Sulphamethoxazole-trimethoprim; W, Trimethoprim; P, Penicillin G; TE, Tetracycline; CAZ, Ceftazidime.

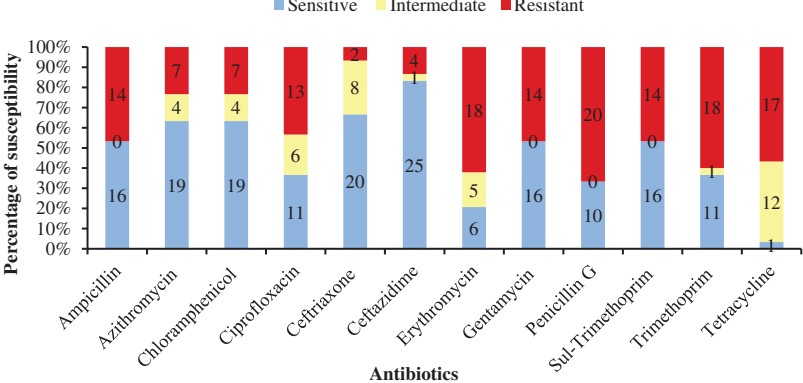

**Figure 4 Antibiotic susceptibility of bacterial isolates.** Percentage of antibiotic susceptibility of representative thirty bacterial isolates in 100% stacked column against 12 antimicrobial agents (Sul-Trimethoprim denotes Sulphamethoxazole-Trimethoprim).

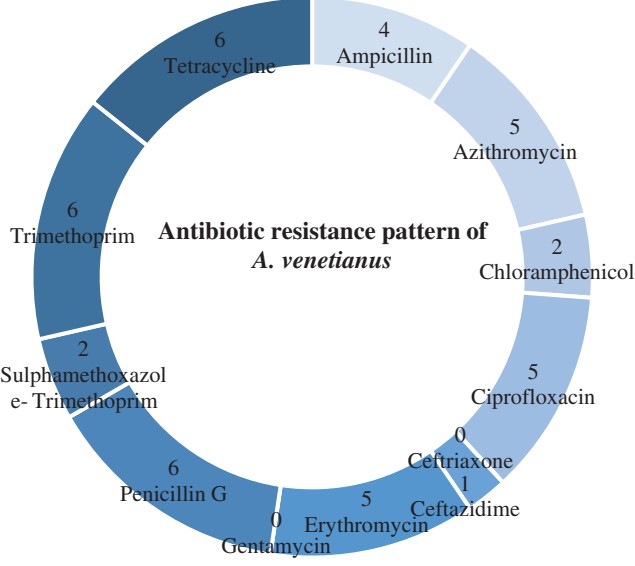

**Figure 5 Antibiotic resistance pattern.** Antibiotic resistance pattern against 12 antimicrobial agents of six studied isolates identified as *Acinetobacter venetianus*.

spp. had MAR index from 0.08 to 0.67. The strain YWO8-97 (*P. monteilii*) had the highest MAR index since the strain was resistant to all tested antibiotics (Fig. 6).

# DISCUSSION

## Identification of diverse bacteria associated with shrimp PL nurseries

The present study found *Vibrio* to be the dominant group of bacteria in the shrimp PL nurseries. It has been revealed that bacterial species belong to the genus *Vibrio* are part of the natural flora of penaeid shrimps (*Gomez-Gil et al., 1998*). *Vibrio* spp. can turn pathogenic, and they can be the causal agents of shrimp mortality in the culture system eventually (*Nash et al., 1992*). Among fifteen identified species of the genus *Vibrio* in the

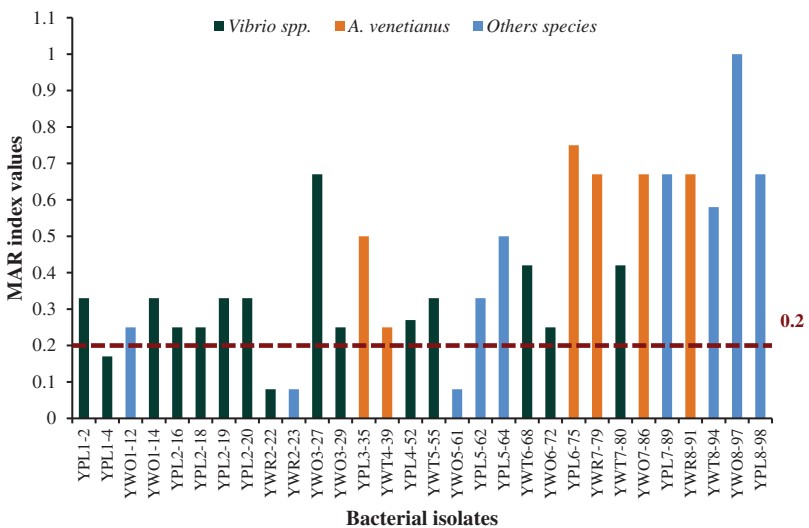

**Figure 6 MAR index.** MAR index values of 30 diversified studied bacterial isolates against 12 anti-microbial agents. Red disconnected line with value (0.2) indicate maximum threshold value of MAR index for a bacterial isolate to be multiple antibiotic resistant.

present study, six isolates were *V. alginolyticus*. *V. alginolyticus* causes vibriosis disease in juvenile penaeid shrimp (*Selvin & Lipton, 2003*). This bacterium has been reported to be isolated from diseased shrimp (*Litopenaeus vannamei*) of culture ponds in Taiwan (*Liu et al., 2004*). In 2019, different shrimp farms of Satkhira district of Bangladesh were suspected to be affected by vibriosis diseases, and phenotypic characterization and 16S rRNA gene sequencing revealed *V. alginolyticus* as the causative agent (*Hannan et al., 2019*). The challenge test for pathogenicity observation found that the identified *V. alginolyticus* caused disease signs in juvenile shrimps and caused high mortality (*Hannan et al., 2019*). Moreover, one of our studied strains was identified as *V. parahaemolyticus*. *V. parahaemolyticus* and other *Vibrio* spp. carrying *pir*A and *pir*B toxin genes in their plasmid are responsible for shrimp disease known as early mortality syndrome/acute hepatopancreatic necrosis disease (EMS/AHPND) in cultured shrimp (*Dong et al., 2017*; *Liu et al., 2018*, *Restrepo et al., 2018*; *Muthukrishnan et al., 2019*) and it caused large scale losses in production of shrimp in Vietnam, China, Malaysia and Thailand (*FAO, 2013*). AHPND positive *V. parahaemolyticus* strain in shrimp farms of Bangladesh have been reported in the past years (*Eshik et al., 2017*; *Eshik et al., 2018*). On the other hand, the present study revealed four strains of *V. shilonii*. Monoclonal antibody test, biochemical tests and 16S rRNA gene sequencing reveals twelve strains of *Vibrio* spp. isolated from shrimp as *V. shilonii* (*Longyant et al., 2008*). Moreover, in the present study the vibrios identified as *V. xuii* and *V. brasiliensis* which were first isolated and described from the marine aquaculture environment (*Thompson et al., 2003*). The pathogenicity study on *Vibrio* spp. to rainbow trout (*Oncorhynchus mykiss*) and *Artemia* nauplii disclosed that *V. brasiliensis* can cause mortality up to 100% (*Austin et al., 2005*). In China, *V. brasiliensis* was reported to be characterized from moribund cultured

Pacific white shrimp in 2021 based on the phylogenetic analysis of 16S rRNA gene and some housekeeping genes, and the researcher reported *V. brasiliensis* as a novel shrimp pathogen (*Li et al., 2021*). In contrast, *V. natriegens* is known as one of the fastest growing nonpathogenic bacteria and a potential organism for many biotechnological researches (*Failmezger et al., 2018*). Bacterial strains isolated from marine soil sediments in Kerala, India have been identified as *V. natriegens* through 16S rRNA gene sequencing and the bacterium has the potential as glycolipid biosurfactant which is antipathogenic against *V. harveyi* to control vibriosis (*Kannan et al., 2019*). Therefore, the *Vibrio* spp. identified from water samples, and shrimp PL samples collected from nurseries in the present study support the findings of the previous studies mentioned. The presence of pathogenic vibrios in PL nurseries can be critical since it may transmit to the shrimp rearing farms, and cause disease when the environmental parameters are in favor of the pathogenic agents.

Our study also found six isolates of *A. venetianus* from tiger shrimp (*P. monodon*) PL nurseries of Khulna and Satkhira districts of Bangladesh. *A. venetianus* is a novel species which was isolated from the sea in Japan, seawater in Israel, oil in Italy and aquaculture ponds in Denmark (*Vaneechoutte et al., 2009*). In China, the red leg disease incidents occurred in almost entire shrimp farming regions of Qingpu, Shanghai in 2017, and as a causative agent *A. venetianus* was confirmed (*Huang et al., 2020*). They stated this as the first report of *A. venetianus* as a potential pathogen of whiteleg shrimp (*L. vannamei*) cultured in freshwater (*Huang et al., 2020*). Additionally, the genus *Acinetobacter* is a potential pathogen in aquaculture which has caused large mortalities in last few years in common carp, channel catfish, Indian major carp, blunt snout bream and silver carp (*Kozinska et al., 2014*; *Cao et al., 2016*; *Behera et al., 2017*; *Cao et al., 2017*; *Malick et al., 2020*). To our knowledge, the finding of the present study is the first report regarding the presence of *A. venetianus* in *P. monodon* PL nurseries of Bangladesh through 16S rRNA gene sequencing. The present study found the presence of this pathogen in the raw sea water, treated water, outlet water, and PL samples of five different nurseries. Therefore, it is a matter of fear regarding *A. venetianus* presence in the shrimp farms of Bangladesh, although it can be considered as an emerging issue to conduct more research.

In our study, three isolates of *S. algae* were identified. This is a rod-shaped marine bacterium which has been isolated from seawater in previous study (*Holt, Gahrn-Hansen & Bruun, 2005*). *S. algae* has been reported as human pathogen (*Satomi, 2014*; *Yousfi et al., 2017*). *S. algae* can also act as an opportunistic pathogen in aquatic animals and in China and Taiwan this bacterium caused abalone mortalities (*Cai et al., 2006*) and ulcerative disease in marine fish, channel bass (*Chang et al., 2003*). Therefore, it is not uncertain to find *S. algae* in shrimp nursery as the present study investigated.

The present study found one strain of *Z. denitrificans* from outlet water of a shrimp nursery. This is a denitrifying, heterotrophic and anaerobic bacterium. *Z. denitrificans* has been previously isolated from the sediments of estuarine mangrove environment in Taiwan along with *Z. taiwanensis*, both have been identified as novel species under a

new genus *Zobellella* (*Lin & Shieh, 2006*). The genome of *Z. denitrificans* (ZD1) is characterized with a four-gene-cluster for which the bacterium can effectively synthesize biodegradable polyhydroxybutyrate (PHB, an alternative to plastics) in salty environment (*Wu et al., 2019*). The reason behind the presence of *Z. denitrificans* in outlet water of PL nursery might be due to the contact of nursery outlet water with the sediment outside which may get salty. There is no report for any infectious incidence due to this bacterial strain.

One strain of *A. caviae* was identified in the present study. From imported shrimp, a study found sixty-three nalidixic acid-resistant *Aeromonas* spp., and phylogenetic analysis of *gyr*B sequences indicated that among those 26 strains were *A. caviae* harboring toxin genes (*Shakir et al., 2012*). In a study of marine shrimp species cultured in low salinity inland ponds, among 87 isolates of *Aeromonas* spp., 7% isolates are *A. caviae* (*Yano et al., 2015*).

In the present study, two isolates were identified as *Pseudomonas* spp. In a study of the Phillippines, 40 bacterial isolates have been differentiated from the tiger shrimp eggs, larvae, PL, the feeds, and the rearing water, and the study observed that the *Vibrio* spp. are dominant group. Moreover, *Vibrio* and *Pseudomonas* spp. were reported to be present in both larvae and rearing water (*Torres, 2007*). Our investigation also found several *Vibrio* spp. and *Pseudomonas* spp. from water and PL samples of shrimp nurseries. In Austria, a total of 520 *Pseudomonas* isolates from different sampling sites of the Danube River have been isolated, and among those, the most of the isolates were *P. putida* and *P. fluorescens* (*Kittinger et al., 2016*). One isolate in our study was identified as *P. monteilii*. In China, *P. monteilii* strains have been isolated from soil samples and identified through 16S rRNA gene sequencing, and it has been suggested through experiment that this particular strain could be useful for bioremediation of contaminated water and soil through the degradation of pyrene (*Ping et al., 2017*). *P. monteilii* has been isolated from the gut of grass carp and had the antibacterial activity against *A. hydrophila* (*Qi et al., 2020*). The previous studies suggest that *P. monteilii* can be beneficial for the environment.

The present study revealed the presence of two Gram-positive bacterial isolates *viz.* *B. licheniformis* and *B. pumilus* in shrimp PL nurseries. *B. licheniformis* is found in plant and soil (*Veith et al., 2004*), and used industrially for manufacturing biochemicals, enzymes, antibiotics, and aminopeptidase (*Rey et al., 2004*). One of the most common probiotic groups is *Bacillus* spp. that is used in aquaculture (*Nayak, 2021*). The growth and survival of shrimp PL without water exchange using marine *B. pumilus* and periphytic microalgae complex is effective in maintaining low levels of total ammonia-nitrogen (TAN) and nitrite-nitrogen ($NO_2$-N), therefore no requirement to change the culture water (*Banerjee et al., 2010*). This complex of bacteria and microalgae also has the potential to increase the PL survival, produce improved quality of shrimp, reduce *Vibrio* counts and therefore, make simpler the larval culture system (*Banerjee et al., 2010*). In our study, we found *B. licheniformis* and *B. pumilus* from raw and outlet water, respectively that could have potential to maintain water quality in PL nurseries and probiotic functions.

## Antibiotic susceptibility pattern and MAR index of studied bacterial strains

In the present study, all 30 bacterial strains isolated from the raw sea water samples, treated water, outlet water and PL samples showed multiple antibiotic susceptibility patterns against 12 different antibiotics (Table 4). The present study found that 26 bacterial strains isolated from shrimp PL nurseries were resistant to three or more antibiotics. This is an indication how antibiotics are being indiscriminately used in the shrimp farms and pose health risk to shrimp and humans by promoting emergence of multidrug-resistant bacterial strains.

The antimicrobial resistance of *Vibrio* spp. isolated from marine whiteleg shrimp farms have a high prevalence of resistance to ampicillin (45.2%) and the tetracycline class (38.7%); same study revealed that multidrug resistance was associated with 29% of *Vibrio* isolates (*Reboucas et al., 2011*). *V. alginolyticus* isolates from oysters in Korea have been identified as resistant to erythromycin and 73.3% was resistant to rifampin (*Kang et al., 2016*).

The previous studies suggested beneficial effects of *P. monteilii*, and this bacterium have anti-pathogenic effect (*Ping et al., 2017*; *Qi et al., 2020*), however the strain identified in our study was resistant to all the tested antibiotic which express high risk of antibiotic usage. On the other hand, the present study identified three strains as *S. algae* and observed that all the strains were susceptible to azithromycin, ceftriaxone and ceftazidime. Most of the *Shewanella* spp. were reported to be susceptible to gentamicin (99%), cefotaxmine (95%), ciprofloxacin (94%), piperacillin and tazobactam (98%) (*Vignier et al., 2013*). However, the present study found three strains that were resistant to six or more antibiotics.

In the present study, 16S rRNA sequencing revealed the presence of *A. venetianus* in the shrimp PL nurseries of southwest regions of Bangladesh. In China whiteleg shrimp, *P. vannamei* has been reported to be affected recently by *A. venetianus* causing red leg disease and the strain showed resistances to different antibiotics with multiple resistances against chloramphenicol, quinolones and tetracyclines (*Huang et al., 2020*). The strains of the present study also showed multiple antibiotic resistance and high resistance prevalence against penicillin G, trimethoprim, tetracycline, azithromycin, ciprofloxacin and erythromycin.

MAR index values ≥0.2 indicates that the bacterial strains were previously exposed to heavy antibacterial contamination, and therefore, are at high risk of multi-drug resistance (*Noorlis et al., 2011*; *Nurhafizah et al., 2021*). The present study found that most of the isolates were resistant to three or more antibiotics having MAR index more than 0.2 including 14 *Vibrio* spp. and six *A. venetianus*. In a study of Ecuadorian *P. vannamei* hatcheries, 20 bacterial strains were isolated from shrimp larvae samples during mortality occurrences and the strains showed similarity to the *Vibrio* sequences as pathogens of the Harveyi clade through the 16S rRNA sequence analysis (*Sotomayor et al., 2019*). Those strains had MAR index ranged from 0.18 to 0.36 (average 0.23) and all the isolates were resistant to penicillin (*Sotomayor et al., 2019*). Our study also found that most of the

studied bacterial strains (68.7%) were resistant to penicillin. *Nurhafizah et al. (2021)* identified three strains of multidrug-resistant *V. harveyi* associated with luminescent vibriosis in *P. vannamei* having MAR index of 0.4. Different species of the genus *Vibrio* from *P. monodon* farms of Bangladesh showed multiple antibiotic resistance (*Rahman et al., 2020*). However, in recent times, antimicrobial resistance has come under the spotlight as the COVID-19 crisis is badly affecting almost the whole world because there are reports of antibiotic usage while treating COVID-19 to avoid any secondary bacterial infections (*Murray, 2020*). Antimicrobial resistance causes about 700,000 deaths per year and it could be deadlier over the time as the prediction indicates that the fatality may rise to 10 million by 2050 (https://foreignpolicy.com/2020/05/21/china-farms-antibiotic-resistance-antimicrobial-amr/). Therefore, the antimicrobial resistance emerges as a global threat, and to fight this battle the World Health Organization (WHO) launched a Global Action Plan (GAP) based on a 'One Health' approach which highlights the interdependence of human health, animal health and the environment (*WHO, 2015*).

## CONCLUSIONS

This study concludes the presence of both nonpathogenic and potentially pathogenic bacteria in shrimp PL and water samples collected from the nurseries of south-west region of Bangladesh. Specifically, there were some bacteria which have been proved in other studies as human pathogen. So, it can be said that there remains risk of bacterial disease while handling them. There were also some bacterial strains in the present study which were proved as pathogenic for shrimp from previous reports, especially *A. venetianus*. The antibiotic susceptibility test proved that there were antibiotic resistant bacteria present in the study sites and most of the isolates showed multidrug resistance. From this study, it may be concluded that shrimp PL nurseries in southern part of Bangladesh are getting contaminated with the potentially pathogenic bacteria either through sea water they are getting from the open sea for rearing the PL or from vertical transmission of broodstock to PL, the feed they used for PL rearing and poor hygiene maintenance. Therefore, it is very important to pay attention to good rearing practice, hygiene, controlled use of antibiotic supplemented feed and overall bio-security measures in the shrimp PL nurseries and rearing farms.

## ACKNOWLEDGEMENTS

The authors wish to express deep gratitude to the shrimp PL nursery owners from where the samples were collected for the investigation. The authors are grateful to the Invent Technologies Ltd., Bangladesh for their kind cooperation during the study.

### Funding

This work was supported by the special allocation research grant of the Ministry of Science and Technology, Bangladesh for 2018-2019 fiscal year. The funders had no role in study design, data collection and analysis, decision to publish, or preparation of the manuscript.

## Grant Disclosures

The following grant information was disclosed by the authors:
Ministry of Science and Technology, Bangladesh for 2018–2019 fiscal year.

## Competing Interests

Mohammad Shamsur Rahman is an Academic Editor for PeerJ.

## Author Contributions

- Abdullah Yasin performed the experiments, analyzed the data, prepared figures and/or tables, authored or reviewed drafts of the paper, and approved the final draft.
- Mst Khadiza Begum conceived and designed the experiments, performed the experiments, analyzed the data, prepared figures and/or tables, authored or reviewed drafts of the paper, and approved the final draft.
- Md. Mostavi Enan Eshik performed the experiments, authored or reviewed drafts of the paper, and approved the final draft.
- Nusrat Jahan Punom performed the experiments, authored or reviewed drafts of the paper, and approved the final draft.
- Shawon Ahmmed performed the experiments, authored or reviewed drafts of the paper, and approved the final draft.
- Mohammad Shamsur Rahman conceived and designed the experiments, performed the experiments, analyzed the data, prepared figures and/or tables, authored or reviewed drafts of the paper, and approved the final draft.

## Field Study Permissions

The following information was supplied relating to field study approvals (*i.e.*, approving body and any reference numbers):

Samples were collected from commercial shrimp PL nurseries with verbal permission from the authority with the promise that the farms' identity will not be disclosed/published.

## Data Availability

The 16S rRNA sequences are available at GenBank: MT368010 to MT368038.

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
