# Peer review of "Molecular identification and antibiotic resistance patterns of diverse bacteria associated with shrimp PL nurseries of Bangladesh: suspecting Acinetobacter venetianus as future threat"

_PeerJ, doi:10.7717/peerj.12808_

## Round 0.1 · original submission · Major Revisions

Our reviewers have identified many points where improvements are needed. Please address their points thoroughly.

Reviewer 1 ·

Basic reporting

-The article title is overwhelmed. The study did not show the risk analysis on Acinetobacter venetianus properly. It is based on author's opinion. Most of contents show the bacterial findings using molecular and microbiology methods. I suggest the authors adjust the title.
- The introduction part is redundant, for example a background on hatcheries was described on line 59 -85. It can be concluded within 2-3 sentences. It should relate to the article topic.
-Result and discussion contents are repetitive in table, figure, and descriptive contents. Contents (bacterial isolation, molecular identification, and antibiotics susceptibility) are redundant which can summarize in tables. Because this study was local finding, discussion part should more precisely on comparison to other regions and reveal readers gain advantage how necessary to read and understand a situation in Bangladesh.
-Overall, of this article, lots of contents are unnecessary. The authors should analyze and summarize which readers can take advantage and do not waste a time to read.

Experimental design

- The sampling method does not show statistical methods. How to make readers confidently understand that samples were representative of PL-hatcheries in this study.
- To give reasonably for risk analysis, statistical analysis on Acinetobacter venetianus should be provided, not based on opinion.

Validity of the findings

- This study is local information, but it could be fundamental data comparing to other geographical regions or industries. Authors can provide more comparative information that be useful for readers.

Additional comments

no comment.

Reviewer 2 ·

Basic reporting

The English writing needs substantial improvement. Standard wording and international numbering units should be used. Refrain from using 'lakh', 'crore' and 'ghers' on their own. Some sentences are unclear and needs to be rephrased. Please observe the consistency, eg., Gram positive/ negative vs gram positive/ negative; rod-shaped vs rod shaped

Additional literature refences are needed as to whether Vibrio should still be regarded as opportunistic pathogens after the emergence of acute hepatopancreatic necrosis disease (AHPND) involving a number of Vibrio spp besides V. parahaemolyticus.

Table 1 needs revision. The colony size should be in mm (diameter) instead of small, medium or large. The gel lanes in Fig 2A need proper labeling.

Experimental design

L146: Live PL?
L150: What tool/ equipment was used for the homogenization?
L159-160: Unclear description...'Only from treated water, 100 μl of processed raw sample and 10-2 dilution was inoculated on different agar plates'. Please revise for better clarity.

There should be negative control for the 16S PCR.

Please add a few more (instead of one or two) homologous 16S sequences from GenBank for each inferred species for multiple seq alignment and construction of phylo tree.

Validity of the findings

Conclusion
It should not be generalized based on the previous studies that all strains of certain bacterial species are pathogenic or non-pathogenic. To conclude if the isolates are pathogenic or otherwise, it should be proven by experimental infection on case by case basis.

Additional comments

Shrimp aquaculture does not actually protect the wild shrimp resources from depletion. While farming produces much more shrimp than catching from the seas, shrimp-fishing industry still continues. Also, some shrimp broodstocks are still being caught from the seas for PL production in hatcheries.

The B.licheniformis and B. pumilus isolates might be associated with probiotic use in the nurseries.

Some of the isolates could be the same strain.

Annotated reviews are not available for download in order to protect the identity of reviewers who chose to remain anonymous.

Reviewer 3 ·

Basic reporting

Overall the manuscript is interesting and can be published after the revisions.

Experimental design

Good.

Please explain more on how many replicates of samples (PL, 3 kinds of water) were collected.

Validity of the findings

yes

Additional comments

It will be good to add more discussion on bacteria association with shrimps that were studied in other countries particular in SE Asia/Asia.

English grammar needs to be check thoroughly.

---

## Round 0.2 · Major Revisions

Please address our reviewers' remaining concerns.

Reviewer 2 ·

Basic reporting

The writing still needs improvement to be clear and unambiguous, and also to get rid of typo and spelling errors, as well as the inconsistent format.

L90-91: Please revise the sentence to be more informative by mentioning the species reported (V. owensii and V. harveyi). Please also refer to this reference:
Vibrio parahaemolyticus and Vibrio harveyi causing Acute Hepatopancreatic Necrosis Disease (AHPND) in Penaeus vannamei (Boone, 1931) isolated from Malaysian shrimp ponds (https://doi.org/10.1016/j.aquaculture.2019.734227), which provides more info than Kondo et al (2015).

L177: Provide the reference for interpretation of susceptibility/ resistance.

L380-381: Please also refer to "Virulence properties and pathogenicity of multidrug-resistant Vibrio harveyi associated with luminescent vibriosis in Pacific white shrimp, Penaeus vannamei" (https://doi.org/10.1016/j.jip.2021.107594) for more relevant interpretation of MAR index value.

Please refer to the enclosed PDF for highlight and annotation.

Experimental design

Method description needs improvement.

Please refer to the enclosed PDF for highlight and annotation.

Validity of the findings

Conclusion:
The bacterial isolates in the present study have not been tested to be pathogenic. The pathogenicity status should not be based solely on the findings of previous studies by other researchers. “Potentially pathogenic” is more appropriate.

Please refer to the enclosed PDF for highlight and annotation.

Additional comments

L41-42: more than 0.2 means prior high exposure to the antibiotics
L61: “theshrimp” typo
L65: nauplii
L82: cause
L116: (after filtration and disinfection)
L120: used as pooled samples
L129: Vibrio spp.
L132: diverse
L133: μl inconsistent with μL
L135: "first processed water sample before dilution" - do you mean "undiluted". Please rephrase for clarity.
L152: Wizard SV Gel and PCR Clean-Up System
L159-160: “The reported sequences in this study have been deposited in the NCBI GenBank database under accession numbers MT368010 to MT368038.” should be in the results

L169: the bacterial isolates have not been tested to be pathogenic yet at this stage, so "pathogenic bacteria" is inappropriate. It should be "the isolates"

L175: "broth culture" instead of "bacterial suspension"
L174: Mueller
L194: estimated 1500 bp
Comment: DNA ladder only estimates the size of DNA band
L212: "100% identity" is more appropriate versus "were confirmed"
Comment: Further in-depth analysis by multilocus sequence analysis (MLSA) may sometimes dispute the identity inferred from 100% similarity shown by BLAST pairwise alignment

L217-218: “The identified bacterial strains have been summarized in the Table 3 according to the water and shrimp PL samples source collected from eight different shrimp PL nurseries.” - this should be placed after the phylogenetic results.

L221: “relatively short sequence” instead of “partial sequence”
Comment: Even the estimated 1500 bp sequences are not considered complete sequences.

L235: Be consistent with the term: antibiotic vs antimicrobial vs antibacterial

L239-240: Rephrase “Tetracycline showed high resistance since there was no sensitive isolates against tetracycline except V. xuii ”
No isolates were sensitive to tetracycline except V. xuii

L246: means prior high exposure to the antibiotics
L247: including
L253: The present study found Vibrio to be
L255: Revise “can turns into opportunistic pathogens” to “can turn pathogenic”
L262: revealed
L264: “caused” instead of “formed”

L266-267: “V. parahaemolyticus is responsible for shrimp disease known as early mortality syndrome/acute hepatopancreatic necrosis disease (EMS/AHPND) in cultured shrimp”…this is not generally true. Please revise.
AHPND is caused by Vp strains and other Vibrio spp that carry pir toxin genes (in plasmid). It should not be generalized that Vp causes AHPND.

L271: revealed
L293: incidents
L307: Revise “which was isolated from seawater” - which has been isolated from seawater in previous study
L312: found
L313: . Z. denitrificans has been previously isolated from...
L314: both have been identified as novel species

L328: observed
L329: were reported to be present
L332: Danube River
L357-358: Revise “This is an indication that how discriminately antibiotics are being used”
This is an indication how antibiotics are being indiscriminately used

L358-359: Revise “and imposing health risk to shrimp and as well as to human by making multidrug-resistant bacterial strains.”
- and pose health risk to shrimp and humans by promoting emergence of multidrug-resistant bacterial strains

L365: suggested
L366: Revise “this species identified” - the strain identified
L369: Most of the Shewanella spp. were
L371-372: found three S. algae strains that were resistant
L373: revealed
L376: showed
L377: showed
L378: high resistance prevalence against
L382: resistant to
L385: Vibrio species not identified?
L389: showed
L390: Revise "has been come to the spotlight" - has come under the spotlight
L390-391: Please clarify how the antimicrobial resistance has come under the spotlight as the COVID-19 crisis is badly affecting almost the whole world

Revise “This study concludes the possible presence of both pathogenic and nonpathogenic bacteria”

Consider: This study concludes the presence of both nonpathogenic and potentially pathogenic bacteria…

L407: potentially pathogenic

Annotated reviews are not available for download in order to protect the identity of reviewers who chose to remain anonymous.

---

## Round 0.3 · accepted · Accept

I am glad to accept your manuscript for publication in PeerJ